# Early childhood parent-reported speech problems in small and large for gestational age term-born and preterm-born infants: a cohort study

Gabrielle Jee ,[1] Sarah Joanne Kotecha,[2] Mallinath Chakraborty ,[2,3] Sailesh Kotecha ,[2,3] David Odd [3,4]

¹Department of Paediatrics, University of Wales Hospital, Cardiff, UK
²Department of Child Health, Cardiff University School of Medicine, Cardiff, UK
³Department of Neonatology, University of Wales Hospital, Cardiff, UK
⁴Department of Population Health, Cardiff University School of Medicine, Cardiff, UK

**Correspondence to**
Dr David Odd;
OddD@cardiff.ac.uk

## ABSTRACT

**Objective** (1) To assess if preterm and term small for gestational age (SGA) or large for gestational age (LGA) infants have more parent-reported speech problems in early childhood compared with infants with birth weights appropriate for gestational age (AGA). (2) To assess if preterm and term SGA and LGA infants have more parent-reported learning, behavioural, hearing, movement and hand problems in early childhood compared with AGA infants.

**Design** Cohort study.

**Setting** Wales, UK.

**Participants** 7004 children with neurodevelopmental outcomes from the Respiratory and Neurological Outcomes of Children Born Preterm Study which enrolled 7129 children, born from 23 weeks of gestation onwards, to mothers aged 18–50 years of age were included in the analysis.

**Outcome measures** Parent-reported single-answer questionnaires were completed in 2013 to assess early childhood neurodevelopmental outcomes. The primary outcome was parent-reported speech problems in early childhood adjusted for clinical and demographic confounders in SGA and LGA infants compared with AGA infants. Secondary outcomes measured were parent-reported early childhood learning, behavioural, hearing, movement and hand problems.

**Results** Median age at the time of study was 5 years, range 2–10 years. Although the adjusted OR was 1.19 (0.92 to 1.55) for SGA infants and OR 1.11 (0.88 to 1.41) for LGA infants, this failed to reach statistical significance that these subgroups were more likely to have parent-reported speech problems in early childhood compared with AGA infants. This study also found parent-reported evidence suggestive of potential learning difficulties in early childhood (OR 1.51 (1.13 to 2.02)) and behavioural problems (OR 1.35 (1.01 to 1.79)) in SGA infants.

**Conclusion** This study of 7004 infants in Wales suggests that infants born SGA or LGA likely do not have higher risks of parent-reported speech problems in early childhood compared with infants born AGA. To further ascertain this finding, studies with wider population coverage and longer-term follow-up would be needed.

## STRENGTHS AND LIMITATIONS OF THIS STUDY

⇒ This study has a relatively large sample size, although this represents around 26% of all eligible invitees to Respiratory and Neurological Outcomes of Children Born Preterm Study (n=26 722).
⇒ Major domains of early childhood developmental outcomes were explored.
⇒ Potentially using parent-reported questionnaires may introduce some degree of recall bias.
⇒ Like most cohort studies, missing data in those enrolled, particularly in confounders, were present.
⇒ Data were collected between 2003 and 2011.

## INTRODUCTION

### Background

Birth weight is a complex product of intrinsic and extrinsic influences on the feto-maternal interaction and is an important predictor of perinatal morbidity and mortality.[1–3] Population studies have shown trends that infants are being born heavier. In England and Wales, there was an average increase of 40 g over 26 years for all live births recorded between 1986 and 2012 with a 8%–10% increased risk of being born with a high birth weight during this period.[4] Trends for heavier birth weights were also observed in other countries including Canada, USA and Sweden, and the cause for this shift is not understood.[4] This drives a need to better understand predictors of birth weight and its association with longer-term outcomes.

Outcomes for certain subgroups such as prematurity and small for gestational age (SGA) have been well explored but remains unclear for large for gestational age (LGA) infants particularly mid-term to long-term outcomes. LGA is associated with birth complications and indirect effects persisting into adulthood.[5] Around birth, larger term infants have increased risk of shoulder dystocia, meconium aspiration, lower 5 min

Apgar scores and death.[6 7] Postnatally, they are more likely to have polycythemia, hypoglycaemia and respiratory distress syndrome; all conditions associated with poor long-term outcomes.[6–8] In preterm births, the prognosis for being LGA is unclear, with some literature suggesting an advantage; reporting lower perinatal mortality in preterm LGA infants, but with higher risks of early-onset sepsis and intraventricular haemorrhage.[9]

Overall, data on the mid-term to long-term effects of being LGA across all gestational ages are lacking, and little is known about neurodevelopmental outcomes. A retrospective cohort study by Moore *et al* found an increased risk of autism in term infants born SGA between 23 and 31 weeks, whereas being born large may have conferred some protective effect.[10] In view of speech being a dynamic product of higher function cognitive and sensorimotor feedback processes involving multiple cortical and subcortical areas for planning, selecting, sequencing and motor programming, it was selected as the primary outcome of interest. Due to the complex interaction between neurodevelopmental domains incuding speech, this study also evaluated learning, behavioural, hearing, movement and hand problems.[11]

### Objectives

1. To assess if preterm and term SGA or LGA infants have more parent-reported speech problems in early childhood compared with infants with birth weights appropriate for gestational age (AGA).
2. To assess if preterm and term SGA and LGA infants have more parent-reported learning, behavioural, hearing, movement and hand problems in early childhood compared with AGA infants.

## METHODS
### Study design
This study was conducted using data collected from the Respiratory and Neurological Outcomes of Children Born Preterm study (RANOPS), a cross-sectional population study conducted in Wales in 2013. This study recruited equal numbers of preterm (n=13 361) and term-born children (n=13 361) in years 2003, 2005, 2007, 2009 and 2011 to complete questionnaires on respiratory and neurodevelopmental outcomes if their child has ever had any active or resolved problems (online supplemental files 1 and 2).

Term infants were selected to be comparable with the preterm infants, for date of birth, sex and locality. A total of 7129 responses were received including consent for data usage and access to health databases. Characteristics between those who enrolled and those that did not are shown in online supplemental file 3. Parents with preterm infants were more likely to respond to the questionnaire than those with term births (p<0.001) and responders were less likely (39.5% vs 53.8%) to live in the most deprived half of Wales. However, the proportions of males (p=0.52) and those with low,

normal or high birth weights for their gestation were similar (p=0.61).

Parent-reported answers on neurodevelopmental outcomes were collected in 2013 for all ages. For example, 'Does your child have any problems with their speech?'; followed by a yes, or no, option. In the event neither option was selected, the response was recorded as unsure. Baseline demographics, including birth and maternal characteristics, were collected from national health databases.[11–13]

The primary neurodevelopmental outcome was parent-reported early childhood speech problems.[11] Secondary neurodevelopmental outcomes were parent-reported early childhood learning, behavioural, hearing, movement and hand problems. For this study, all 'unsure' responses were re-coded as 'no' and included in the primary analysis.

### Study population
The eligible population were all children enrolled in RANOPS, born from 23 weeks of gestation to mothers from 18 to 50 years of age with available speech outcomes (n=7004). Exposure measures were gestational age at birth and birthweight centile by category. Birthweight centiles were calculated for each sex and gestation (in weeks) using the LMS Growth programme (Medical Research Council).[14] SGA was defined as <10th centile on the UK-WHO growth charts and >90th centile for LGA. These are values generally accepted across England and Wales, with Scotland using the 5th and 95th centiles.[15 16] Gestational age was categorised (as per WHO definitions) as extremely preterm (<28 weeks), very preterm (28–31 weeks) and moderate to late preterm (32–<37 weeks).[17] Post-maturity was defined as gestational age at birth greater than or equal to 42 weeks.[18] SGA and LGA infants were compared with AGA infants across gestations.

### Covariates
Covariates included neonatal and maternal influences known a priori to influence birth weight in-utero. Neonatal factors are singleton or multiple births, gestational age at birth and sex. Maternal factors accounted for, included smoking during pregnancy, the Welsh Index of Multiple Deprivation (WIMD) score (a measurement of relative deprivation for small areas with scores of 1–1909, 1 being the most deprived) and their age.[19] These overlap with potential influences on the primary outcome, parent-reported speech problems in early childhood. While complex and multifactorial in potentially influencing the primary outcome, maternal socioeconomic status was accounted for using WIMD scores.[20] Predictors of low birth weight include low birth weight,[21] multiple pregnancies[22] and foetal sex.[23] Mode of delivery was not assumed to impact birth weight and considered likely multifactorial due to perinatal clinical practice surrounding estimated birth weight or centile, birth complications and maternal preference.[15 24 25] The determinants of high birth weight are, however, less clear.

**Table 1** Neonatal and maternal birth characteristics for all including preterm-born and term-born infants

| Characteristics | n | Birthweight centile | | | P value |
| --- | --- | --- | --- | --- | --- |
| | | SGA (<10th) | AGA (10th–90th) | LGA (>90th) | |
| Age at survey (median years) | 7003 | 4.08 (2.67–7.67) | 5.17 (2.75–7.67) | 4.37 (2.75–7.67) | 0.69 |
| *Neonatal factors* | | | | | |
| Male | 7004 | 313 (48.68%) | 2458 (45.50%) | 450 (46.92%) | 0.26 |
| Singleton | 7004 | 507 (78.85%) | 4647 (86.02%) | 904 (94.26%) | <0.001 |
| Gestation (median weeks) | 7004 | 35 (33–38) | 36 (34–39) | 36 (34–38) | <0.001 |
| Preterm (<37 weeks) | 7004 | 458 (71.23%) | 3105 (57.48%) | 623 (64.96%) | <0.001 |
| Post-term (≥42 weeks) | 7004 | 25 (3.89%) | 99 (1.83%) | 5 (0.52%) | <0.001 |
| Mode of delivery | 4091 | | | | <0.001 |
| Unassisted vaginal delivery | 2037 | 119 (32.16%) | 1659 (52.33%) | 259 (47.01%) | |
| Instrumental | 406 | 27 (7.30%) | 334 (10.54%) | 45 (8.17%) | |
| Elective caesarean section | 511 | 66 (17.84%) | 349 (11.01%) | 96 (17.42%) | |
| Emergency caesarean section | 1137 | 158 (42.70%) | 828 (26.12%) | 151 (27.40%) | |
| *Maternal factors* | | | | | |
| Age (mean years)* | 6225 | 29.85 (6.00) | 30.35 (5.66) | 31.15 (5.37) | <0.001 |
| WIMD score (median)† | 6804 | 845 (428–1351) | 985 (503–1447) | 985 (524–1444) | <0.001 |
| WIMD decile† | 6804 | 5 (3–7) | 6 (3–8) | 6 (3–8) | <0.001 |
| Maternal smoking‡ | 6725 | 133 (21.38%) | 655 (12.65%) | 75 (8.11%) | <0.001 |

Values are n (%), mean (SD) or median (IQR) as appropriate.
Denominator between measures differs due to missing data.
*Maternal age at the time of delivery.
†Welsh Index of Multiple Deprivation/Deciles of WIMD (lower values reflecting more deprivation).
‡Maternal smoking during pregnancy.
AGA, appropriate for gestational age; LGA, large for gestational age; SGA, small for gestational age; WIMD, Welsh Index of Multiple Deprivation.

Evidence suggests that active smoking is associated with increased risk of low birth weight and preterm birth, and prevalence of smoking is associated with socioeconomic status.[26 27] There is also an increased risk of low birth weight with younger mothers, perhaps influenced by socioeconomic status. However, evidence surrounding birth weight and advanced maternal age is inconsistent, with some studies showing increased prevalence of infants born LGA, while others showing a U-shaped trend of low birth weight with increasing age.[26 28–31]

### Statistical analysis

Duplicates and infants with missing exposure or primary outcome data were removed. Initially, neonatal and maternal characteristics at birth for all infants were compared across their birthweight centile category. Comparisons were performed using the Kruskal-Wallis equality of populations ranks test for birthweight centile category, gestational age and WIMD score at birth. Sex, number of births, mode of delivery and maternal smoking during pregnancy were compared using the $\chi^2$ test and maternal age at delivery using the analysis of variance test.

Next, the proportion of children with speech problems between birthweight centile categories was compared. Using a logistic regression model, the unadjusted and adjusted for potential confounders, ORs for speech problems, comparing SGA and LGA infants to those born AGA were derived. Stratifying by preterm and term infants and using a logistic regression model, the ORs for speech problems were also derived for SGA and LGA infants compared with AGA infants. In view of a single primary outcome, p values corresponding to each statistical test were not corrected.

Six sensitivity analyses were performed. First, we repeated the logistic regression using a random-effects model to cluster by week of gestational age, and second by the age of the child at time of the survey. We then repeated using the 5th and 95th centile cut-offs, and then again removing all responders where 'unsure' was coded for the primary outcome.[16] The main analysis was also repeated with an ordinal logistic regression analysis looking at the odds of an increasing number of reported neurodevelopmental disorders. Finally, the analysis was repeated using a missing data technique (Multiple Imputation with Chain Equations (details in online supplemental file 4)) to assess the impact of missing outcome and covariate data on the association seen.[17] Likelihood ratio tests were used to compare models. Analysis was performed using Stata SE V.17 (Statacorp LLC).

### Patient and public involvement

Patients or the public were not involved in the design, conduct, reporting, or dissemination plans of this research.

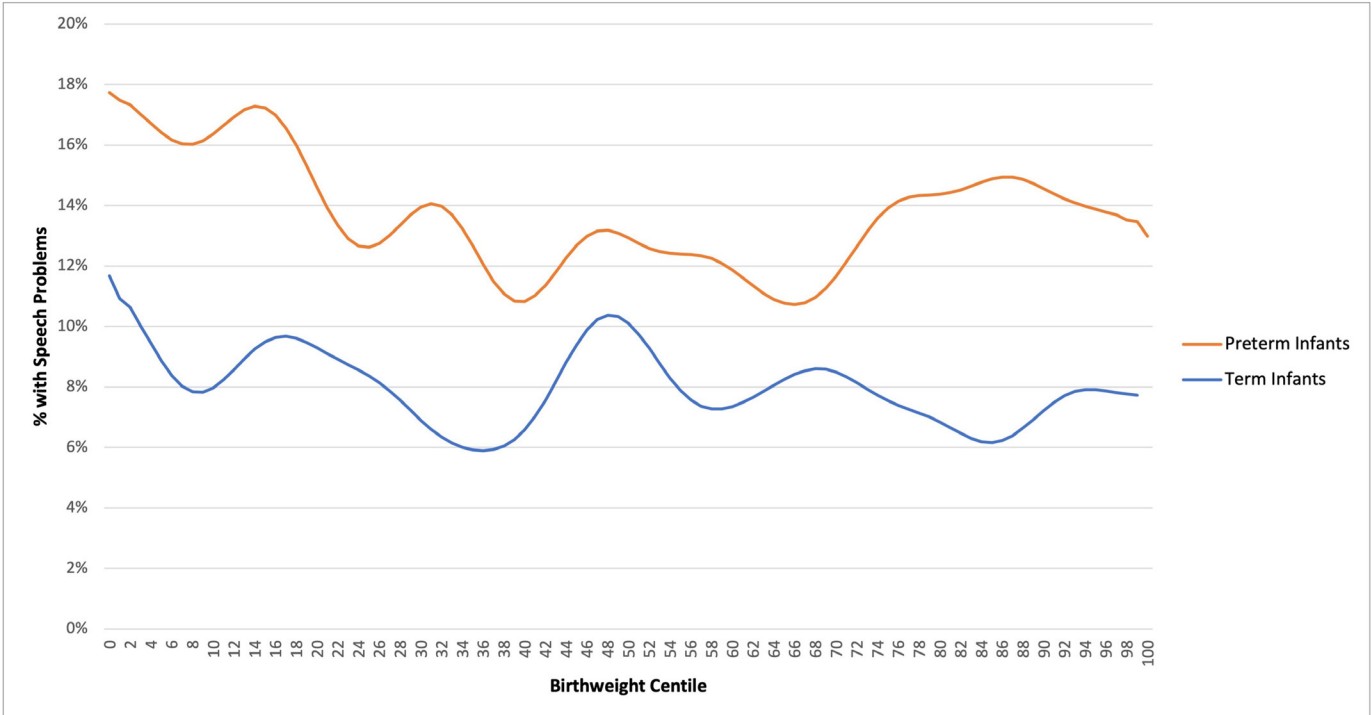

**Figure 1** Proportion of preterm and term infants with parent-reported speech problems in early childhood by birthweight centile (Gaussian smoothed).

## RESULTS

Four thousand two hundred and eighty-four preterm and 2865 term infants were enrolled in RANOPS (n=7149).[12] Twenty duplicates were removed, leaving 7129 children. Those born to mothers younger than 18 or above 50 years of age were excluded (n=83), leaving 7046 eligible children. Eight infants had missing birthweight centiles, and a further 34 had missing data on the primary outcome, early childhood speech problems, leaving a study population of 7004 participants for the primary analysis (99.40% of the eligible responders).

Stratifying by birthweight centile categories of SGA, AGA and LGA infants, baseline neonatal and maternal characteristics of the study cohort are shown in table 1. The median age at time of the survey was 4.08–5.17 (p=0.69). Besides the distribution of sex across birthweight centile categories, there were significant differences in other baseline neonatal and maternal characteristics analysed.

Distribution of early childhood speech problems in preterm and term infants by birthweight centile is as shown in figure 1. Figure 2 depicts the number of neurodevelopmental disorders stratified by gestation age at birth, birthweight centile category and deprivation measures or WIMD category (quintile). Analysis of the distribution of the primary outcome, parent-reported speech problems showed a significant difference between different birthweight centile categories, p=0.05. There were significant differences between birthweight centile categories and the secondary outcomes, except for parent-reported hearing problems (p=0.59) (table 2).

Logistic regression adjusted for neonatal and maternal features of the primary outcome (parent-reported speech problems in early childhood) showed that SGA and LGA infants were not more likely than their AGA counterparts to have problems, OR 1.19 (0.92 to 1.55) and OR 1.11 (0.88 to 1.41), respectively. Analysis of parent-reported secondary outcomes found evidence that SGA infants have an increased risk of learning difficulties in early childhood compared with those born AGA, OR 1.51 (1.13 to 2.02). There was also some evidence that SGA infants were more likely to have early childhood behavioural problems, OR 1.35 (1.01 to 1.79). This study did not find that infants born SGA or LGA were more likely to have hearing, movement or hand problems in comparison to AGA infants (table 3).

Analysis was repeated using a random-effects regression model, stratified by preterm-born or term-born, and the adjusted ORs did not demonstrate that preterm infants born SGA or LGA were more likely to have parent-reported speech-problems than infants born AGA. There was also no evidence of interaction between the exposures measured ($P_{interaction}$=0.999) (table 4).

Multilevel regression model of SGA and LGA babies compared with those born AGA across each gestational age category (using child age as the clustering variable) also showed compatible results (SGA, OR 1.19 (0.92 to 1.55); LGA, OR 1.11 (0.87 to 1.41)) with the main analysis. When 'unsure' responses excluded from analysis, logistic regression repeated showed no increased likelihood in primary neurodevelopmental outcome, parent-reported

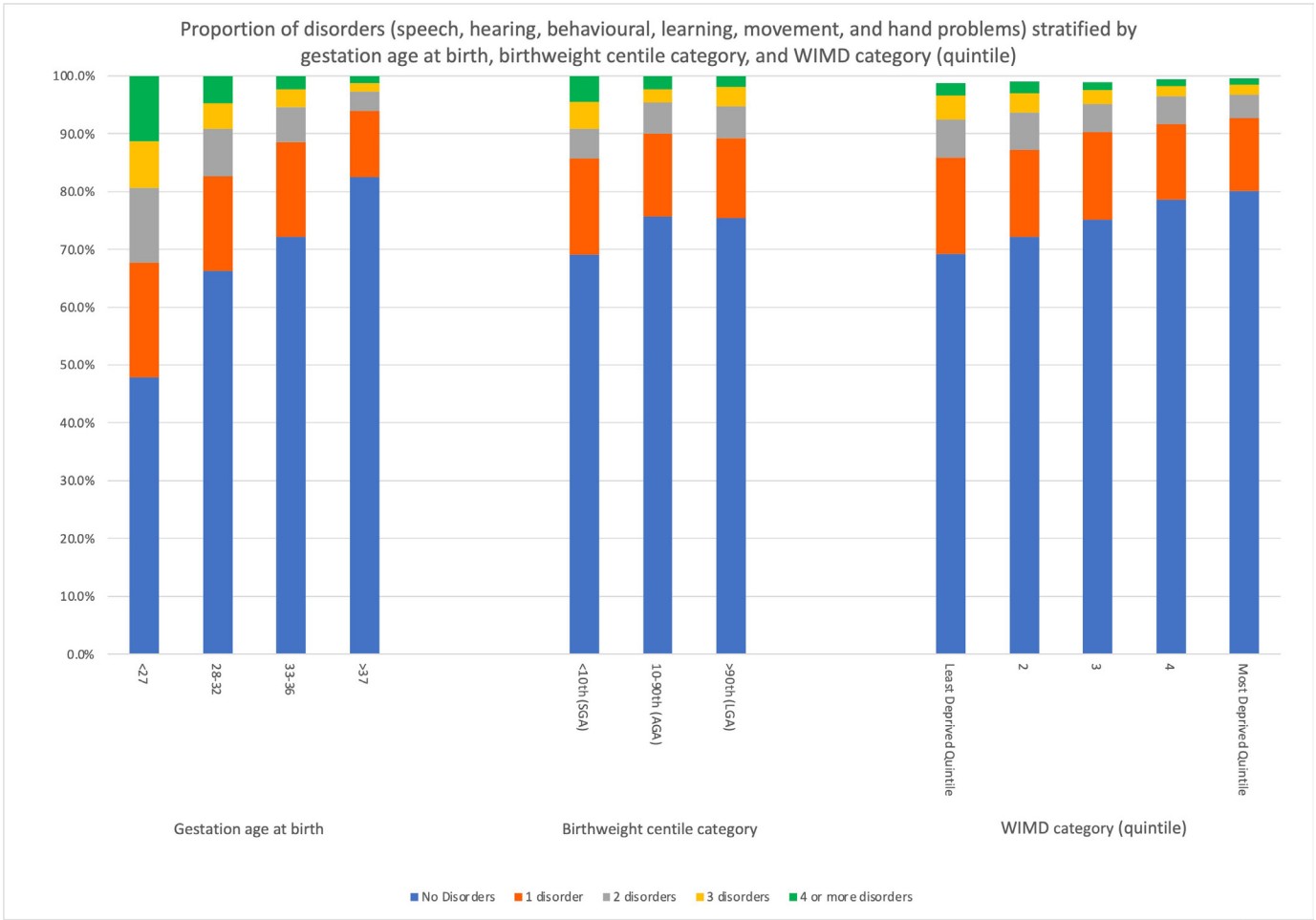

**Figure 2** Proportion of neurodevelopmental disorders stratified by gestation age at birth, birthweight centile category and WIMD category (quintile). AGA, appropriate for gestational age; LGA, large for gestational age; SGA, small for gestational age; WIMD, Welsh Index of Multiple Deprivation.

early childhood speech problems in babies born SGA, OR 1.20 (0.92 to 1.56) or LGA, OR 1.13 (0.90 to 1.44) compared with those born AGA. Repeat of the analysis using the 5th and 95th centiles as cut-offs for SGA showed that babies born SGA were not more likely than babies born AGA to have parent-reported early childhood speech problems, OR 1.35 (0.98 to 1.86).[16] There was

also no evidence to suggest more parent-reported early childhood speech problems in babies born LGA, OR 1.11 (0.84 to 1.48). Repeat of the main analysis examining the odds of an increasing number of reported developmental disorders found results compatible with the primary analysis (SGA, OR 1.17 (0.96 to 1.43); LGA, OR 1.11 (0.93 to 1.32)). Finally, repeating the main analysis

| Table 2 | Early childhood neurodevelopmental outcomes across all birthweight categories | | | | |
|---|---|---|---|---|---|
| | | **Birthweight centile** | | | |
| **Outcome measure** | **n** | **SGA (<10th), n (%)** | **AGA (10th–90th), n (%)** | **LGA (>90th), n (%)** | **P value** |
| Primary | | | | | |
| Speech problems | 7004 | 93 (14.46) | 603 (11.16) | 112 (11.68) | 0.05 |
| Secondary | | | | | |
| Learning difficulties | 6980 | 83 (12.95) | 404 (7.50) | 70 (7.34) | <0.001 |
| Behavioural problems | 6963 | 86 (13.46) | 459 (8.54) | 77 (8.11) | <0.001 |
| Hearing problems | 6985 | 49 (7.66) | 371 (6.89) | 73 (7.62) | 0.59 |
| Movement problems | 6990 | 45 (7.01) | 240 (4.45) | 49 (5.12) | 0.01 |
| Hand problems | 7000 | 40 (6.22) | 206 (3.82) | 36 (3.76) | 0.01 |
| AGA, appropriate for gestational age; LGA, large for gestational age; SGA, small for gestational age. | | | | | |

**Table 3** Unadjusted and adjusted ORs for early childhood neurodevelopmental outcomes in SGA and LGA infants compared with AGA controls

| Outcome measures | Unadjusted | | | Adjusted for neonatal features* | | | Adjusted for neonatal* and maternal features† | | |
| | | Birthweight centile | | | Birthweight centile | | | Birthweight centile | |
| | n | SGA (<10th) | LGA (>90th) | n | SGA (<10th) | LGA (>90th) | n | SGA (<10th) | LGA (>90th) |
|---|---|---|---|---|---|---|---|---|---|
| **Primary** | | | | | | | | | |
| Speech problems | 7004 | 1.34 (1.06 to 1.70) | 1.05 (0.85 to 1.30) | 7004 | 1.28 (1.01 to 1.63) | 1.01 (0.82 to 1.26) | 5935 | 1.19 (0.92 to 1.55) | 1.11 (0.88 to 1.41) |
| **Secondary** | | | | | | | | | |
| Learning difficulties | 6980 | 1.83 (1.42 to 2.36) | 0.98 (0.75 to 1.27) | 6980 | 1.71 (1.32 to 2.22) | 0.92 (0.71 to 1.21) | 5894 | 1.51 (1.13 to 2.02) | 1.06 (0.79 to 1.41) |
| Behavioural problems | 6963 | 1.67 (1.30 to 2.13) | 0.95 (0.74 to 1.22) | 6963 | 1.57 (1.22 to 2.02) | 0.89 (0.69 to 1.15) | 6963 | 1.35 (1.01 to 1.79) | 1.19 (0.91 to 1.57) |
| Hearing problems | 6985 | 1.12 (0.82 to 1.53) | 1.12 (0.86 to 1.45) | 6985 | 1.06 (0.77 to 1.45) | 1.08 (0.83 to 1.40) | 5900 | 1.03 (0.73 to 1.47) | 1.16 (0.87 to 1.54) |
| Movement problems | 6990 | 1.62 (1.16 to 2.25) | 1.16 (0.84 to 1.59) | 6990 | 1.39 (0.99 to 1.94) | 1.08 (0.78 to 1.49) | 5901 | 1.43 (0.98 to 2.09) | 1.31 (0.92 to 1.84) |
| Hand problems | 7000 | 1.67 (1.18 to 2.37) | 0.98 (0.69 to 1.41) | 7000 | 1.56 (1.09 to 2.23) | 0.95 (0.66 to 1.36) | 5910 | 1.36 (0.90 to 2.06) | 1.11 (0.75 to 1.64) |

*Adjusted for foetal sex, singleton or multiple birth and gestational age at birth.
†Adjusted for maternal age and WIMD score at the time of birth and maternal smoking status during pregnancy.
AGA, appropriate for gestational age; LGA, large for gestational age; SGA, small for gestational age; WIMD, Welsh Index of Multiple Deprivation.

using a multiple imputation model (n=7004) (SGA, OR 1.21 (0.95 to 1.54); LGA, OR 1.06 (0.85 to 1.32)) was also compatible, with no clear associations seen.

## DISCUSSION

The literature supports that parents' perceptions on their child's development have shown significant consistency with standardised developmental objective measures in preterm-born and term-born infants, and are increasingly used in developmental screening for at risk infants.[32–40] Example of such is the validated PARCA-R (Parent Report of Children's Abilities) for very-preterm infants at 2 years.[41] Although some studies have reported variation between parental estimates and objective measures of

speech development, parent-reported speech outcomes especially speech intelligibility still remains widely used in clinical practice.[42–44] While the absolute numbers reported here are large, they do only represent a relatively low proportion of those invited to enrol in the study; although they had similar low or high birthweigths compared with those who did not enrol. However, they also appeared to come from less deprived areas than the wider population, and interpretation of our findings should consider this.

The production of speech is complex, relying on not only intact cognitive, motor and sensory functions but also complicated by hearing loss, particularly the child's age at time of hearing loss.[32 45] Reassuringly, in this work,

**Table 4** Unadjusted and adjusted ORs for primary neurodevelopmental outcome of early childhood speech problems stratified by preterm or term in SGA and LGA infants compared with AGA controls

| Gestation | Birthweight category | Unadjusted (n=7004) | | Adjusted for neonatal features* (n=7004) | | Adjusted for neonatal* and maternal features† (n=5935) | |
| | | OR (95% CI) | P$_{interaction}$ | OR (95% CI) | P$_{interaction}$ | OR (95% CI) | P$_{interaction}$ |
|---|---|---|---|---|---|---|---|
| Preterm (<37 weeks) | SGA (<10th) | 1.27 (0.98 to 1.66) | 0.864 | 1.30 (0.99 to 1.71) | 0.844 | 1.19 (0.88 to 1.61) | 0.999 |
| | LGA (>90th) | 1.04 (0.81 to 1.33) | | 1.05 (0.82 to 1.35) | | 1.12 (0.85 to 1.47) | |
| Term (≥37 weeks) | SGA (<10th) | 1.16 (0.69 to 1.96) | | 1.21 (0.71 to 2.05) | | 1.19 (0.69 to 2.07) | |
| | LGA (>90th) | 0.92 (0.60 to 1.42) | | 0.91 (0.59 to 1.41) | | 1.10 (0.71 to 1.72) | |

*Adjusted for foetal sex, singleton or multiple birth.
†Adjusted for maternal age and WIMD score at time of birth and maternal smoking status during pregnancy.
AGA, appropriate for gestational age; LGA, large for gestational age; SGA, small for gestational age; WIMD, Welsh Index of Multiple Deprivation.

we were unable to identify clear associations between SGA, or LGA, infants and adverse parent-reported speech outcomes in early childhood; in either the unadjusted or adjusted models, as well as in the analysis of hearing problems. Also reassuringly, this study also did not find an increased risk of adverse parent-reported behavioural and hand problems in SGA and LGA infants, movement problems in SGA infants and learning difficulties in LGA infants. However, CIs are relatively wide and important increases in morbidity cannot be excluded and more work with precise estimate may be warranted.

Evidence from this study suggest that SGA infants may have higher risks of parent-reported learning difficulties and behavioural problems in early childhood but not hearing, movement or hand problems. A single-centre cohort study of term infants in Australia born to women presenting antenatally between 1981 and 1984 showed that children born SGA had significantly more learning difficulties when followed up at 14 years of age using a parent-reported survey and academic achievement test. This study reported a comparable 19.8% and 20.5% incidence for those who completed psychometric testing and behavioural questionnaires, respectively. This study also demonstrated long-term attention difficulties in extremely SGA (3rd centile and below) term-born female adolescents.[46] Similar findings of poorer attention, executive function and memory were reported in a small cohort study of SGA preterm-born and term-born young adults who underwent neuropsychological assessment.[47] Similarly, a cross-sectional study of 5181 childrens' behaviour, between ages 4 and 15 years in England, using a validated parent-reported questionnaire and stratifying for sociodemographic factors, suggested an association between birth weight and behavioural problems in children.[6 48]

We also saw that the proportion of all parent-reported early childhood disorders showed a strong relationship with gestational age at birth (seen in figure 2). This appears consistent with the wider literature,[49–51] and this work was designed to adjusted, in part, for this by using birthweight centiles. Alternatively, this may, in part, reflect the complexities and co-dependency of many of these outcomes as gestational age lowers, and further work is currently underway to look at the phenotypes and interactions of challenges these infants demonstrate.

Analysis of baseline characteristics noted a significant difference in maternal age, and demographics at birth between birthweight centile categories. While this study was not designed to explore the relationship between maternal age and birthweight centile categories, previous studies have reported significant findings of higher maternal age in LGA infants.[28–31] In addition, the association between maternal smoking, deprivation and birthweight centile categories was sobering, with 20% of SGA infants exposed to in-utero smoking. In this work, more than 1 in 5 mothers smoked during pregnancy, identifying an important public health focus on a modifiable risk factor for low birth weight to address,[26] with even first trimester cessation of smoking having substantial benefits.[26 52 53]

There was also a significant difference in the mode of delivery across birthweight centile categories likely due to clinical practice managing high-risk pregnancies.[15 24 25] This may include singleton or multiple pregnancies, gestational age and estimated birth weight or centile at time of birth. Sex was not associated with SGA or LGA, compatible with findings of a large cohort study in Netherlands.[23]

## CONCLUSIONS

Findings from this work, on a subset of 7004 infants in Wales suggest that infants born SGA or LGA may not have higher risks of speech problems in early childhood when compared with AGA infants. While enrolment was achieved on only a subset of less deprived infants, important differences may still exist, and we found that some infants being born SGA may have increased parent-reported learning difficulties and behavioural problems compared with AGA infants. Further longer-term studies on infants born SGA and LGA would be of value to better understand the association of birth weight on neurodevelopment.

**Acknowledgements** We would like to thank all participants in this study.

**Contributors** GJ conceived and designed this work, wrote statistical analysis plan, cleaned and analysed data, drafted the first manuscript, revised draft manuscript and edited and updated the final paper. DO conceived and designed this work, wrote statistical analysis plan, analysed data and drafted the first manuscript, revised draft manuscript and edited and updated the final paper. DO is also responsible for the overall content as guarantor. SJK and SK conceived, designed, performed original RANOPS including data collection, revised draft manuscript and edited and updated the final paper. MC revised the draft manuscript and edited and updated the final paper.

**Funding** No additional funding was received for this analysis. The original study RANOPS was funded by Medical Research Council Experimental Medicine Challenge Grant (reg: MR/M022552/1), Mason Medical Research Foundation grant and Children and Young Peoples Research Network Wales.

**Competing interests** None declared.

**Patient and public involvement** Patients and/or the public were not involved in the design, or conduct, or reporting, or dissemination plans of this research.

**Patient consent for publication** Not applicable.

**Ethics approval** Ethical approval was sought at initiation of RANOPS and approved by South East Wales Research Ethics Committee (Research Ethics Committee 12/WA/0155 Project 91349). Parents provided written consent to participate.

**Provenance and peer review** Not commissioned; externally peer reviewed.

**Data availability statement** Data are available upon reasonable request. All data must be held securely at Cardiff University as per ethical approval given for this research. Anonymous data will be available from the Department of Child Health at Cardiff University to bonafide researchers provided that ethical approval is obtained from a research ethics committee in the UK for any suggested studies. Requests for data access should be sent to SK (Kotechas@cardiff.ac.uk).

**ORCID iDs**
Gabrielle Jee http://orcid.org/0000-0001-6647-1110
Mallinath Chakraborty http://orcid.org/0000-0002-1721-6532
Sailesh Kotecha http://orcid.org/0000-0003-3535-7627
David Odd http://orcid.org/0000-0002-6416-4966

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
