## [Reviewer comments · BMJ Open]

ARTICLE DETAILS

TITLE (PROVISIONAL)	Early childhood parent-reported speech problems in small and large for gestational age term- and preterm-born infants: A cohort study
AUTHORS	Jee, Gabrielle; Kotecha, Sarah; Chakraborty, Mallinath; Kotecha, Sailesh; Odd, David

VERSION 1 – REVIEW

REVIEWER	Rahman, Amanda Staten Island University Hospital
REVIEW RETURNED	02-Nov-2022

GENERAL COMMENTS	I commend the authors for further investigating the relationship between infant birth weight and neurodevelopmental outcomes. As the authors discuss, fetal and neonatal growth is known to affect not only neonatal outcomes but also the growth and outcome of the later infant and child. The authors attempt to correct for additional covariates that are relevant to the topic at hand and have a large sample size. Although this topic is relevant and an important one, I do think that before this manuscript is ready for publication, a few areas must be addressed: Speech development is an important neurodevelopmental outcome. However, the authors use "speech problems" as an independent marker of complex neurocognitive function. While speech is one part of neurocognition, there are many components and the selection of speech development on its own is incomplete. The standard for assessment of neurocognition would be a formal neurocognitive outcome or assessment. Due to the limitations of the data collection, this is not available. It is more appropriate therefore to provide the data as it is and evaluate simply birthweight and speech problems in early childhood. The definitions of the outcomes are very vague. While it is true this is survey-based data, disclosure of the specific questions being asked, or more description of the actual marker itself, is necessary. Was it simply, "Does the child have speech problems?". There are several challenges to this ambiguity: 1. Large variation in interpretation of the question: The authors mention recall bias, but do not comment on differences in interpretation. Speech problems could vary from a stutter or lisp to complete lack of expressive language and these obviously have very different clinical implications. The same is true of the other variables. Was any subjective data collected?2. Variation in child's age at time of survey: This is shown by the large variation in median age. Again, the authors do mention recall bias, but do not discuss the timing of the outcomes in question.
--

	That is, were parents asked about current problems? Problems ever? Problems at a certain age? In addition to recall bias, this creates challenges as many older children may have outgrown a speech delay or learning problem, and may therefore be considered "normal" per parent survey. In addition to these larger concerns, there are a few smaller comments that will provide more clarity:  1. More consistent usage of terminology would be helpful. The authors appropriately define and use the terminology SGA/LGA but also use other phrases when discussing SGA/LGA/AGA (low birthweight, appropriate birthweight). While this is technically correct, consistent semantics (unless referencing another resource with a different definition) would be helpful in ensuring consistent understanding. For brevity, I recommend consistent use of SGA, LGA, and AGA. 2. In the abstract, since there was no acute intervention, it would be more useful to combine the "Interventions" and "Main Outcome measures" both into the "Outcome measures". it would also be useful to include mention of the secondary outcomes evaluated, as results are listed later. 3. In the results, there is repetition between the data presented in the text and tables. It would be more useful to describe the pertinent findings in the text, or data that is not available in the tables. However, data in the tables does not need to be replicated in the text. 4. In the tables, it would provide more clarity to label the columns as SGA, AGA, and LGA. I wish the authors the best of luck with this important work, and hope that with some changes and clarity, it can be used to further our understanding of the longer term impacts of birthweight on outcomes.
--	---

REVIEWER	Kim, Hyun Min University of Otago, Biostatistics, Dean's Department
REVIEW RETURNED	16-Nov-2022

GENERAL COMMENTS	This study is aimed at addressing important research objectives looking at the childhood developmental impacts of small for gestational age (SGA) and large for gestational age (LGA) births. The study used data from a reasonably sized cohort (n=7004 to n=3542) to analyse the association between parent perceived speech problems (and other developmental problems) and SGA/AGA/LGA births adjusted for some background variables mainly related to birthweight and prematurity. The manuscript is well written and clearly describes the analyses performed. However, there are some major concerns regarding the application of particular outcome measures and analysis to address the stated research objectives. Hence the conclusions reached are not adequately supported by the results presented in the manuscript. Overall, some recommendations and feedback for this manuscript are:  1) Use of precise and specific terminologies throughout the manuscript: e.g., title should clearly state that it is a study about SGA and LGA rather than birthweight itself. The period for study outcomes is "childhood" or "2-7 years" rather than "long-term". The outcome is "parent perceived problems" not actual, directly measured speech or neurocognitive function problems.
--

	2) Parent reported or parent perceived problems used as the main outcome variable here are not assessed for sensitivity, consistency or validity. There are well-known childhood neurodevelopmental instruments (e.g., Bayley Scales (BSID)) or screening tools (SDQs) and tests for language development (e.g., CELF, BPVS, PPVS) that would be more suitable for assessing or reporting developmental problems. However, parent perceived problems are important for screening purposes and can be useful for early detection of potential problems for further diagnostic testing. There also may be important differences between parents' perception of child development between Preterm and Term-born cohorts. The questionnaire described in this study seems to be customised for the cohort. Some rationale and explanation would be helpful to demonstrate the reliability or validity of this measure. Currently, the chosen outcome measure is not suitable for addressing the research objectives. 3) The results for each of the parent-perceived problems are reported. However, the rationale for using speech problems as a surrogate marker for all other high-level neurocognitive function is not adequately described nor supported by prior research. Language/speech development is an important domain of neurodevelopment and cognitive function, but it is not often applied as an overall measure of development. A more layered and nuanced approach is recommended to provide a better overall picture of a child's development. Even with the current measures used, one can describe the children who had multitude of problems in 6 different domains – e.g., children with 4 or greater number of parent concerns for development and describe the distribution of this composite against other variables such as gestational age, birthweight, SGA/AGA/LGA categories, deprivation. 4) A large proportion (~50%) of cohort was missing for the adjusted models reported in Table 3. The analysis, discussion and limitations sections do not adequately describe the potential impact of this, why they may be missing adjustment variables, or what were the major differences between groups who were missing and were not missing in these analyses. 5) The choice of confounding factors needs to be explained in terms of speech/language development and not just the exposure variable. Given that many social factors influence language development, a possible impact of intermediate factors such as parent-child interaction and early childhood education are also important factors to consider. 6) The study objectives seem to be around examining the impact of birthweight (over and beyond that of prematurity or gestational age) on child development. It is important to acknowledge this distinction from studies looking at the impact of prematurity or very low birth weight since a baby who is born very preterm may still have appropriate birthweight for their gestational age. Such categorisation is likely to mask the true impact of difficult births (very preterm and very low birthweight births).
--	--

VERSION 1 – AUTHOR RESPONSE

Responses to Reviewer 1, Dr. Amanda Rahman's comments:

I commend the authors for further investigating the relationship between infant birthweight and neurodevelopmental outcomes. The authors attempt to correct for additional covariates that are relevant to the topic at hand and have a large sample size.

- We would like to thank the reviewer for this appreciating comment.

1. Speech development is an important neurodevelopmental outcome. However, the authors use "speech problems" as an independent marker of complex neurocognitive function. While speech is one part of neurocognition, there are many components and the selection of speech development on its own is incomplete. The standard for assessment of neurocognition would be a formal neurocognitive outcome or assessment. Due to the limitations of the data collection, this is not available. It is more appropriate therefore to provide the data as it is and evaluate simply birthweight and speech problems in early childhood.

Answer: We agree that neurocognition encompasses several domains including speech. We apologise as it was not made clear through our title and objective and have amended this. Our aim was to evaluate each of the six domains in small (SGA) and large for gestational age (LGA) infants compared to appropriate for gestational age (AGA) infants with speech as our primary outcome of interest due to easier quantification by parents and reflects the child's ability to perform a higher-level cognitive task.

2. Large variation in interpretation of the question: The authors mention recall bias, but do not comment on differences in interpretation. Speech problems could vary from a stutter or lisp to complete lack of expressive language and these obviously have very different clinical implications. The same is true of the other variables. Was any subjective data collected?

Answer: Again, apologies for the lack of clarity. We have attached a copy of the questionnaire as a supplementary material to aid readers' understanding of the questions asked. Unfortunately, we do not have sufficient data to provide detailed breakdowns of parent-reported problems for each child, and have made this clearer in the limitations section of the discussion

3. Variation in child's age at time of survey: This is shown by the large variation in median age. Again, the authors do mention recall bias, but do not discuss the timing of the outcomes in question. That is, were parents asked about current problems? Problems ever? Problems at a certain age? In addition to recall bias, this creates challenges as many older children may have outgrown a speech delay or learning problem, and may therefore be considered "normal" per parent survey.

Answer: We have amended our methods highlight that this study accounts for all problems experienced by the child from birth to time of questionnaire, describing an example of a question in the parent-reported questionnaire. As aforementioned, we have included the questionnaire as a supplementary material. We have also added an additional sensitivity analysis using the child's age as a random effect, to adjust for a diagnostic bias across the age of the child. Interpretation remains identical.

4. More consistent usage of terminology would be helpful. The authors appropriately define and use the terminology SGA/LGA but also use other phrases when discussing SGA/LGA/AGA (low birthweight, appropriate birthweight). While this is technically correct, consistent semantics (unless referencing another resource with a different definition) would be helpful in ensuring consistent understanding. For brevity, I recommend consistent use of SGA, LGA, and AGA.

Answer: Thank you for highlighting this. We have made changes throughout the manuscript.

5. In the abstract, since there was no acute intervention, it would be more useful to combine the "Interventions" and "Main Outcome measures" both into the "Outcome measures". It would also be useful to include mention of the secondary outcomes evaluated, as results are listed later.

Answer: We have merged as recommended, and included the secondary outcomes evaluated.

6. In the results, there is repetition between the data presented in the text and tables. It would be more useful to describe the pertinent findings in the text, or data that is not available in the tables. However, data in the tables does not need to be replicated in the text.

Answer: We have removed replicated data in the text as recommended.

7. In the tables, it would provide more clarity to label the columns as SGA, AGA, and LGA.

Answer: We appreciate this point and have made amendments.

Responses to Reviewer 2, Dr. Hyun Min Kim's comments:

The manuscript is well written and clearly describes the analyses performed.

- This comment is much appreciated and thank you.

1. Use of precise and specific terminologies throughout the manuscript: e.g., title should clearly state that it is a study about SGA and LGA rather than birthweight itself. The period for study outcomes is "childhood" or "2-7 years" rather than "long-term". The outcome is "parent perceived problems" not actual, directly measured speech or neurocognitive function problems.

Answer: We have made changes throughout the manuscript to address this inconsistency by using SGA, LGA, and AGA, and parent-reported. We have also defined the period for study outcomes as "early childhood".

2. Parent reported or parent perceived problems used as the main outcome variable here are not assessed for sensitivity, consistency or validity. There are well-known childhood neurodevelopmental instruments (e.g., Bayley Scales (BSID)) or screening tools (SDQs) and tests for language development (e.g., CELF, BPVS, PPVS) that would be more suitable for assessing or reporting developmental problems. However, parent perceived problems are important for screening purposes and can be useful for early detection of potential problems for further diagnostic testing. There also may be important differences between parents' perception of child development between Preterm and Term-born cohorts. The questionnaire described in this study seems to be customised for the cohort. Some rationale and explanation would be helpful to demonstrate the reliability or

validity of this measure. Currently, the chosen outcome measure is not suitable for addressing the research objectives.

Answer: We appreciate the comment and take onboard the concerns raised. We have amended our discussion to reflect the rationale and evidence for the usage of parent-reported outcomes in this study.

3. The rationale for using speech problems as a surrogate marker for all other high-level neurocognitive function is not adequately described nor supported by prior research. Language/speech development is an important domain of neurodevelopment and cognitive function, but it is not often applied as an overall measure of development. A more layered and nuanced approach is recommended to provide a better overall picture of a child's development. Even with the current measures used, one can describe the children who had multitude of problems in 6 different domains – e.g., children with 4 or greater number of parent concerns for development and describe the distribution of this composite against other variables such as gestational age, birthweight, SGA/AGA/LGA categories, deprivation.

Answer: Speech was chosen as the primary outcome measure for this paper for reasons of speech being a higher function product of complex motor and sensory functions, and the relative ease of quantification by parents. However, we agree that reporting a broader picture of the child's development would strengthen the paper and have added a figure with the number of disorders seen in each child; split by gestation, birthweight centile and deprivation measures. In addition, the main analysis has been repeated (a 6th sensitivity analysis) with an ordinal logistic regression analysis looking at the odds of an increasing number of reported developmental disorders (results are compatible with the primary analysis).

4. A large proportion (~50%) of cohort was missing for the adjusted models reported in Table 3. The analysis, discussion and limitations sections do not adequately describe the potential impact of this, why they may be missing adjustment variables, or what were the major differences between groups who were missing and were not missing in these analyses.

Answer: We have repeated the analysis using a missing data technique (Multiple Imputation with Chain Equations) to assess the impact of missing outcome and covariate data on the association observed. This has been detailed in our results, with the model details uploaded as a supplement document. The results remained essentially the same.

5. The choice of confounding factors needs to be explained in terms of speech/language development and not just the exposure variable. Given that many social factors influence language development, a possible impact of intermediate factors such as parent-child interaction and early childhood education are also important factors to consider.

Answer: Thank you for highlighting this. We have amended our manuscript to account for the impact of covariates such as maternal socioeconomic status on speech outcomes. We agree that it would be useful to include influences occurring after birth such as parent-child interaction and early childhood education. Unfortunately, this data is not available.

6. The study objectives seem to be around examining the impact of birthweight (over and beyond that of prematurity or gestational age) on child development. It is important to acknowledge this distinction from studies looking at the impact of prematurity or very low birth weight since a baby who is born very preterm may still have appropriate birthweight for their gestational age. Such categorization is likely to mask the true impact of difficult births (very preterm and very low birthweight births).

Answer: We again apologise for the suboptimal clarity, and have amended the title, objective, and manuscript as appropriate to reflect and acknowledge this distinction.

VERSION 2 – REVIEW

REVIEWER	Kim, Hyun Min University of Otago, Biostatistics, Dean's Department
REVIEW RETURNED	24-Jan-2023

GENERAL COMMENTS	Overall Comment: The authors have made some substantial revisions in the current manuscript addressing many of the concerns raised in the previous review. However some important issues remain. In particular, the conclusions reached are too definitive for the study design. Although the main findings are supported by extensive sensitivity analyses, all results suggest a consistent direction in the estimated effects for SGA (vs AGA) and lower confidence limits close to 1. This needs to be considered in relation to findings from other studies - how confident are we in making generalisations from this cohort/study? Were these groups reasonably representative of these populations? Perhaps, the concluding statement can be qualified to apply to the studied cohort or suggest further studies replicating these findings in other populations. Overall, some recommendations and feedback for this manuscript are:  1) The revised title is more precise for this study. The period of the study outcomes however was updated inconsistently. Throughout the manuscript, the outcomes are considered “long-term”. There are many studies with longer term follow up over 20 years now. I would consider revising this to “childhood” or early to mid-childhood throughout the manuscript. 2) The primary outcome measure is now revised to “parent reported or parent perceived problems”. 3) The proportion of children with a multitude of problems was explored using Figure 2. This figure is informative and it would be helpful to have brief discussion on the observed pattern, for example, why there seem to be greater differences according to the gestational age but not the birthweight centiles? Also check the deprivation labels as the most deprived seem to have less problems. This may be so but seems slightly counter-intuitive given the summary statistics in Table 1. 4) A large proportion (~50%) of cohort was missing for the adjusted models reported in Table 3. The missingness seems to
--

	be predominantly driven by the “mode of delivery” variable. In the Methods section under “Covariates” (p9 lines 3-6), the “mode of delivery was not assumed to impact birthweight” yet it had been included in the multiple regression models in Table 3. (there is more discussion on the mode of delivery in the last paragraph of “Discussion” – that it was different across birthweight centiles) Unless the mode of delivery had some notable impact on the fit of the model or was considered relevant to the multiple regression model for other reasons, it seems to reduce the degrees of freedom available for model fitting without conferring much benefit. Hence this may negatively impact on efficiency even further – we can see quite a big change in the margin of error from the unadjusted model to the adjusted models in Table 3. 5) Sensitivity Analyses: Various sensitivity analyses have shown similar results to the multiple regression models in Table 3. The effect sizes based on the point estimates seem reasonable (OR around 1.3 for SGA vs AGA), however, the lower confidence limit on most of these are close to 1 suggesting more conservative results than the current statement made in the “Conclusions” section. 6) Conclusions: Given that the outcomes are based on parent-perceived problems, the statement about the “risks of speech problems” (first sentence under Conclusions) needs to be made more specific to the outcome. Also, following up on the comment above in 5), perhaps recommend further studies/research replicating the main findings of this research in other populations (especially given the large evidence base around poorer outcomes among children born preterm). Specific Comments: Abstract Page 3. Conclusions: suggest qualifying the statement, please see above comments. Page 4. Strengths and limitations: lines 12-15 “25% of the possible eligible population” contradicts the study participants section in the Results. Page 6. Lines 19-22: “relative ease of quantification by parents”. There were no quantification of problems in this study – the outcome measure was binary Yes or No. OBJECTIVES: Lines 40-42: The secondary outcomes were also parent-reported. Please revise in this instance as not clear from the context here. METHODS Page 7. Study Design Lines 19-24: Free text responses were not analysed in this study – please consider removing. Page 9. Covariates: Lines 3-6: Please clarify the inclusion of the mode of delivery variable in the adjusted models. RESULTS: Page 12: Lines 5-6: Please clarify “99.4% of the eligible population” this contradicts the statement above in Abstract. DISCUSSION: Page 16: Line 5. “developmental screening for at risk” – a word missing?
--	---

VERSION 2 – AUTHOR RESPONSE

Responses to Reviewer 2, Dr. Hyun Min Kim comments:

In particularly, the conclusions reached are too definitive for this study design. Although the main findings are supported by extensive sensitivity analyses, all results suggest a consistent direction in the estimated effects for SGA (vs AGA) and lower confidence limits close to 1. This needs to be considered in relation to findings from other studies – how confident are we in making generalisations from this cohort/study? Were these groups reasonably representative of these populations? Perhaps, the concluding statement can be qualified to apply to the studied cohort or suggest further studies replicating these findings in other populations.

Answer: We appreciate this fresh insight in helping us revisit our findings and eventual conclusion as although this is one of the largest infant cohort study in Wales, this does represent about 26% of all invitees to the study. Thus, we have highlighted this in our strengths and limitations and amended our conclusion to apply to this studied cohort alongside recommendations for future studies to enable this potential generalisation. We have added a comparison of the known data between those who were and were not enrolled and reflected on this in the manuscript.

1. The revised title is more precise for this study. The period of the study outcomes however was updated inconsistently. Throughout the manuscript, the outcomes were considered long-term. There are many studies with longer term follow up over 20 years now. I would consider revising this to “childhood” or early to mid-childhood throughout the manuscript.

Answer: We have revised the manuscript to read “early childhood” throughout.

2. The primary outcome measure is now revised to “parent reported or parent perceived problems”
3. The proportion of children with a multitude of problems was explored using Figure 2. This figure is informative and it would be helpful to have brief discussion on the observed pattern, for example why there seem to be greater differences according to the gestation age but not birthweight centiles? Also check the deprivation labels as the most deprived seem to have the less problems. This may be so but seems slightly counter-intuitive given the summary statistics in Table 1.

Answer: Apologies for the error; we have amended the deprivation labels for Figure 2. We have revised our manuscript to include a paragraph discussing the observed pattern in Figure 2 as detailed in paragraph 4 in the “Discussion” section.

4. A large proportion (~50%) of the cohort was missing for the adjusted models reported in Table 3. The missingness seems to be predominantly driven by the “mode of delivery” variable. In the Methods section under “Covariates” (p9 lines 3-6), the “mode of delivery was not assumed to impact birthweight” yet it had been included in the multiple regression models in Table 3. (there is more discussion on the mode of delivery in the last paragraph of “Discussion” – that it was different across birthweight centiles). Unless the mode of delivery had some notable impact on the fit of the model or was considered relevant to the multiple regression model for other reasons, it seems to reduce the degrees of freedom available for model fitting without conferring much benefit. Hence, this may negatively impact on efficiency even further – we can see quite a big change in the margin of error from the unadjusted model to the adjusted models in Table 3.

Answer: We have revisited this and omitted “mode of delivery” in our models as aforementioned that this is assumed to not affect birthweight, and our results have been updated as appropriate to reflect this. No change in interpretation was seen.

5. Sensitivity analyses: various sensitivity analyses have shown similar results to the multiple regression models in Table 3. The effect sizes based on the point estimates seem reasonable (OR around 1.3 for SGA vs AGA), however, the lower confidence limits on most of these are close to 1, suggesting more conservative results than the current statement made in the “Conclusions” section.

Answer: We have revised our conclusion as aforementioned.

6. Conclusions: Given that the outcomes are based on parent-perceived problems, the statement about “risks of speech problems” (first sentence under Conclusions) needs to be made more specific to the outcome. Also, following up on the comments in 5) perhaps recommend further studies/research replicating the main findings of this research in other populations (especially given the large evidence base around poorer outcomes among children born preterm).

Answer: As aforementioned, we have revised our conclusion to reflect findings in our study population of 7004 infants in Wales and have suggested further studies to explore possible generalisation of this finding to other populations.

7. Specific comments:

- a. Abstract, page 3, conclusions: suggest qualifying the statement.
Answer: We have amended our conclusion as recommended.

- b. Abstract, page 4, strengths and limitations: lines 12-15 “25% of the possible eligible population” contradicts the study participants sections in the results.
Answer: Apologies that we have not made this sufficiently clear and have revised this statement to include the denominator or all infants invited to participate in RANOPs, n=26722, to aid reader’s understanding and derivation of this percentage.

- c. Page 6, lines 19-22: “relative ease of quantification by parents”. There were no quantification of problems in this study – the outcome measure was binary Yes or No.
Answer: We have removed this from our manuscript.
- d. Objectives, lines 40-42: the secondary outcomes were also parent-reported. Please revise this instance as not clear from the context here.
Answer: This has been revised.
- e. Page 7, methods, study design lines 19-24: free text responses were not analysed in this study – please consider removing.
Answer: This has been removed as well suggested.
- f. Page 9, covariates, lines 3-6: please clarify the inclusion of the mode of delivery variable in the adjusted model.
Answer: Mode of delivery has been removed from our models as detailed in our response above.
- g. Page 12, lines 5-6: please clarify “99.4% of the eligible population” this contradicts the statement above in Abstract.
Answer: We have revised our statement this 99.4% to emphasise is the final eligible study population, n=7004, out of all responders whereas the statement above in “abstract” correlates to 26% derived from all infants invited to participate in RANOPs.
- h. Discussion, page 16, line 5: “developmental screening for at risk” – a word missing?
Answer: We have amended this sentence.
- i. Conclusions: please see my comments above.
Answer: As very well suggested and much appreciated, we have amended our conclusions to include the comments and recommendations.

VERSION 3 – REVIEW

REVIEWER	Kim, Hyun Min University of Otago, Biostatistics, Dean's Department
REVIEW RETURNED	03-Apr-2023
GENERAL COMMENTS	Thank you for the opportunity to review this manuscript. All previous concerns are addressed in the amended manuscript. I have no further comments.